# The Therapeutic Potential of Psychedelics in Treating Substance Use Disorders: A Review of Clinical Trials

**DOI:** 10.3390/medicina61020278

**Published:** 2025-02-06

**Authors:** Lavinia Hogea, Dana Cătălina Tabugan, Iuliana Costea, Oana Albai, Laura Nussbaum, Adriana Cojocaru, Leonardo Corsaro, Teodora Anghel

**Affiliations:** 1Neuroscience Department, “Victor Babes” University of Medicine and Pharmacy, 300041 Timisoara, Romania; hogea.lavinia@umft.ro (L.H.); nussbaum.laura@umft.ro (L.N.); adriana.cojocaru@umft.ro (A.C.); anghel.teodora@umft.ro (T.A.); 2Neuropsychology and Behavioral Medicine Center, “Victor Babes” University of Medicine and Pharmacy, 300041 Timisoara, Romania; 3Psychology Department, West Univerity of Timisoara, 300223 Timisoara, Romania; 4Internal Medicine Department, “Victor Babes” University of Medicine and Pharmacy, 300041 Timisoara, Romania; albai.oana@umft.ro; 5Campus Bio Medico, 00128 Roma, Italy; leo_corsaro@hotmail.it

**Keywords:** psychedelic-assisted therapy, addiction treatment, drug–psychotherapy combination, lysergic acid diethylamide, MDMA, psilocybin, ayahuasca

## Abstract

*Background and Objectives*: Substance use disorders (SUDs) affect millions worldwide. Despite increasing drug use, treatment options remain limited. Psychedelic-assisted therapy (PAT), integrating psychedelic substances with psychotherapy, offers a promising alternative by addressing underlying neural mechanisms. *Materials and Methods*: This review’s purpose is to investigate the current understanding of psychedelic therapy for treating SUDs, including tobacco, alcohol, and drug addiction. The systematic review approach focused on clinical trials and randomized controlled trials conducted from 2013 to 2023. The search was performed using PubMed, Google Scholar, and Consensus AI, following PRISMA guidelines. Studies involving psychedelics like LSD, psilocybin, ibogaine, and ayahuasca for treating various addictions were included, excluding naturalistic studies and reviews. *Results*: Our results highlight the key findings from 16 clinical trials investigating psychedelic therapy for SUDs. Psychedelics like psilocybin and ayahuasca showed promise in reducing alcohol and tobacco dependence, with psilocybin being particularly effective in decreasing cravings and promoting long-term abstinence. The studies revealed significant improvements in substance use reduction, especially when combined with psychotherapy. However, the variability in dosages and study design calls for more standardized approaches. These findings emphasize the potential of psychedelics in SUD treatment, though further large-scale research is needed to validate these results and develop consistent protocols. *Conclusions*: This research reviewed the past decade’s international experience, emphasizing the growing potential of psychedelic therapy in treating SUDs pertaining to alcohol, tobacco, and cocaine dependence. Psychedelics such as psilocybin and ketamine can reduce cravings and promote psychological well-being, especially when combined with psychotherapy. However, regulatory barriers and specialized clinical training are necessary to integrate these therapies into mainstream addiction treatment safely. Psychedelics offer a promising alternative for those unresponsive to conventional methods.

## 1. Introduction

Addiction, or substance use disorder (SUD), is a global public health interest that affects millions of people every day around the world [1]. Symptoms of SUDs may include compulsive consumption, failed efforts to quit using, cravings, social and work-life problems, tolerance, and withdrawal symptoms [2].

The most recent data from 2022 showed that the global drug user population has risen to 292 million, marking a 20% increase in the past decade. Cannabis is still the most popular drug worldwide, with 228 million users. It is followed by opioids with 60 million users, amphetamines with 30 million users, cocaine with 23 million users, and ecstasy with 20 million users [3]. Also, approximately a quarter of the European adult population smokes, and 8.4% consume alcohol daily, according to a report from 2019 [4,5].

Research indicates that psychedelics can be particularly effective in treating conditions like treatment-resistant depression, anxiety disorders, and substance use disorders. Johnson et al. emphasize the need to understand the mechanisms of action of psilocybin and related compounds, which can lead to long-lasting positive changes in mood and well-being, especially in individuals who have not responded to conventional treatments [6].

Addiction is a chronic disease in which substances are used to increase dopamine levels in various regions of the brain, affecting the brain’s reward circuit. The repeated stimulation of dopamine acts as a positive incentive for drug use and reward. With chronic use, the release of dopamine induced by substances diminishes and reduces the need for reward, resulting in tolerance and the necessity for higher doses to achieve the same level of satisfaction [1,7].

Although drug consumption has risen, treatment access is still limited; only one in eight individuals with SUD receive treatment [8]. The efficacy of treatment relies on a blend of medication and psychotherapy. The primary pharmacological approaches for managing SUD focus on curbing cravings for drugs, reducing or preventing drug tolerance, and alleviating or eliminating the harsh symptoms associated with withdrawal. Substitution therapy might also be employed as an effective risk-reduction method. The available drugs include antagonists or receptor agonists and medications targeting different pharmacological aspects, like enzymes that break down the substance ingested [9,10].

The use of natural substances for their mind-altering properties dates back to ancient times in human history. Records exist of cannabis, various other plants, and hallucinogenic mushrooms being used thousands of years ago. These substances often played a role in the sacred ceremonies of indigenous cultures [11].

We continue to divide psychedelics into two main groups: classic and non-classic. Classic psychedelics are typically agonists of serotonergic receptors: lysergic acid diethylamide (LSD), dimethyltryptamine (DMT), psilocybin, and mescaline [1,12,13].

Non-classic psychedelics are substances that interact with various pharmacological targets. For instance, ketamine serves as an antagonist to the N-methyl-d-aspartate receptor; 3,4-methylenedioxymethamphetamine (MDMA) functions as both a reuptake inhibitor and releaser for serotonin and dopamine; ibogaine is an indole alkaloid that affects multiple molecular targets; salvinorin A activates the kappa-opioid receptor; and tetrahydrocannabinol (THC) is a partial agonist at cannabinoid receptors. Other hallucinogenic substances include atropine and scopolamine, which are cholinergic antagonists, and muscimol, along with ibotenic acid [1,12,13,14].

Research into psychedelics flourished from the 1950s to the 1970s. However, once these substances were designated as highly dangerous in the United States, their use was heavily restricted worldwide. This led to the establishment of political and cultural obstacles that hindered further research into their properties [1,15,16].

Even though psychedelics are often restricted, interest in their therapeutic application is increasing for the treatment of depression, anxiety, post-traumatic stress disorder PTSD, and SUD [11,17,18]. Psychedelics might be a breakthrough in the creation of new treatments because they influence various neurotransmitter systems beyond just the dopaminergic one. Psychedelic-assisted therapy (PAT) consists of ongoing psychotherapy, either individually or in a group setting, with a single psychedelic dose administered during a session specifically tailored for patients to focus on overcoming their SUD [7,18,19].

The increasing focus on research into PAT calls for an updated review of how psychedelic drugs impact individuals with SUD. The socio-cultural, political, and ethical constraints tied to historical, correlational, and naturalistic studies render them suboptimal for such an analysis. This review seeks to consolidate the existing knowledge within the field on the therapeutic effects of psychedelics.

The current understanding of PAT for SUD is encouraging, yet significant gaps persist due to historical barriers rooted in socio-cultural, political, and ethical challenges. While earlier research has shed light on the ability of psychedelics to modulate neurotransmitter systems beyond the dopaminergic pathway, robust evidence on their therapeutic mechanisms, efficacy across different substances, and long-term impacts remains incomplete. This systematic review seeks to bridge these knowledge gaps by synthesizing recent findings and clarifying the practical applications of PAT in SUD treatment. Although similar reviews have addressed related questions, this work is vital in integrating the latest data on these therapies’ safety, efficacy, and implementation. By exploring both classic and non-classic psychedelics, this review delves into their contributions to multi-component interventions, particularly those combining pharmacological treatments with psychotherapy. It also addresses the nuanced needs of patients and the complex contexts in which these therapies are delivered, providing a comprehensive and updated perspective on their therapeutic potential.

In this systematic review, we aim to present the current state of knowledge on PAT, covering various substances and methods of application used for patients suffering from SUD, such as tobacco, alcohol, or other drugs.

The objective of this study is to systematically review current evidence on PAT for treating SUD. It aims to evaluate the effectiveness, safety, and mechanisms of both classic and non-classic psychedelics, as well as their role within multi-component interventions like psychotherapy. By addressing gaps in knowledge and exploring clinical applications, this study seeks to inform future research and improve treatment strategies for individuals with SUD.

## 2. Materials and Methods

### 2.1. Objectives and Search Strategy

This paper presents a systematic review focused on experimental research aimed at exploring the effects of psychedelics on human subjects with different addictions. The goal was to identify and analyze the latest studies within the past decade (2013–2023), following the guidelines provided by the Preferred Reporting Items for Systematic Reviews and Meta-Analyses (PRISMA) protocols [20].

For the literature search, electronic methods were employed to find articles pertinent to the research, and a manual review of references in the chosen publications was conducted. The authors reviewed the findings’ titles and abstracts to assess their relevance based on the inclusion criteria.

Articles were searched using PubMed, Google Scholar, and the AI tool Consensus for the following principal expression: [“psychedelics-assisted therapy in addiction”]. Our research aimed to analyze the latest research on PAT, restricting our search period from 1 January 2013 to 31 December 2023. Two authors reviewed the search results, specifically the titles and abstracts. Each publication deemed potentially relevant during the search was scrutinized for its suitability for this review, with a thorough assessment based on predefined inclusion and exclusion criteria. Eight thousand, seven hundred and thirty-four articles were excluded from the review primarily because they did not align with the study’s focus, such as addressing PAT in contexts other than addiction, or because they were identified as reviews, meta-analyses, non-experimental studies, book chapters, or research protocols.

### 2.2. Selection Standards

Our investigation focused on clinical trials and randomized controlled trials conducted between 2013 and 2023 and published in English. We specifically chose studies that involved psychedelic substances such as LSD, psilocybin, MDMA, and ayahuasca/DMT for treating various addictions, including alcohol, tobacco, and other drugs. We excluded naturalistic and correlational studies, book chapters, review articles, and case reports. The research needed to present unique datasets. We did not include studies on healthy subjects, animal research, reviews, or meta-analyses.

### 2.3. Relevant Data from Studies

Several key details were gathered from the studies, including the lead author, publication year, study location, study design, number of participants, the patients’ types of addiction, the psychedelics used, the dosages given, the target condition, outcome measures, and details of the administration and treatment program setting. In the end, each study’s principal findings and conclusions were documented.

## 3. Results

### 3.1. Study Selection

A total of 10,325 papers were found by searching through databases like PubMed, Google Scholar, and Consensus AI. After eliminating 40 duplicate entries and excluding 8734 articles not aligned with the study’s focus, 1551 articles were selected for further screening. Upon closer examination, an additional 876 articles were excluded, primarily because they addressed PAT in the context of psychiatric disorders other than addiction. We removed articles identified as reviews, meta-analyses, non-experimental studies, book chapters, or research protocols, in line with our inclusion criteria. Ultimately, 16 essential articles from the last decade, consisting of clinical trials and randomized controlled trials specifically investigating the use of PAT in treating addiction, were included in our review. The PRISMA flowchart in Figure 1 illustrates the detailed selection process, ensuring a systematic approach to identifying the most relevant and high-quality studies for our analysis.

### 3.2. Details of Included Studies

The articles included in the study had several key features: year of publication, study design, number of participants, type of addiction studied, psychedelic substance and dosage used, target condition, outcome measures, setting, primary findings, and treatment program details (Table 1). These characteristics were selected to ensure a comprehensive understanding of how PAT is applied in treating SUD.

We focused on the year of publication to analyze recent trends, as research on psychedelics has gained momentum in the last decade. Knowledge of the study design, such as randomized controlled trials (RCTs) or open-label studies, helps assess the quality and reliability of the findings. The number of participants is important for understanding the statistical power of each study. The type of addiction and psychedelic substance used provide insights into which substances are being studied for specific disorders.

Outcome measures, such as changes in substance use and psychological effects, help gauge the treatment’s efficacy. Additionally, setting details (clinical or community-based) provide context on how these therapies are administered. The main findings highlight the effectiveness of the treatment, while treatment program details allow for the comparison of different therapeutic approaches.

This systematic approach helped ensure that only relevant, high-quality studies were included in the analysis of PAT for addiction. In this study, a summary of research exploring the use of psychedelics in addiction treatment is provided. The Table 1 highlights key details, including the first author, publication year, DOI, study design, number of participants, type of dependency studied, psychedelic used (including dosage), and the target condition (Table 1).

### 3.3. Descriptive Analysis of the Studies

We began by performing a descriptive analysis of the articles in this review, closely examining the features of recent studies on psychedelic therapy for SUD patients. There were no restrictions regarding the patients’ countries of origin or where the studies were conducted. Between 2013 and 2023, 11 relevant articles met the research methodology’s inclusion criteria. The median publication year was 2016, highlighting the growing importance and interest in this field within the scientific community. In terms of study designs, the review included four randomized controlled trials [21,22,24,29], three open-label pilot studies [23,26,28], one cross-sectional study [30], two qualitative studies [25,31], and one prospective experimental study [27]. When categorizing by addiction type, the studies covered alcohol dependence (four) [21,22,23,25] and tobacco users trying to quit smoking (four) [26,28,30,31], with one study on illicit drug users [24], one on cocaine addiction [29], and one focusing on mixed dependencies (alcohol, tobacco, and cocaine) [27]. The study sample sizes ranged from 10 to 358 patients, with an average of 55.18 participants, a median of 15, and a standard deviation of 98.58. The studies used various psychedelics, mainly psilocybin, ibogaine, and ayahuasca. Psilocybin was often administered at 25 mg/70 kg, though some studies did not specify amounts [21,22,23,24,26,28,29,31]. Ayahuasca was given in traditional doses without exact measurements [25,27]. One study used noribogaine in increasing concentrations [24]. One study used ketamine at 0.71 mg/kg [29], and another combined psilocybin and LSD without stating dosages [30]. Overall, dosage varied across studies, showing different approaches.

### 3.4. Comparative Analysis of PAT for SUD

Each study had a different focus, but most evaluated substance use (alcohol, smoking, or cocaine), cravings, and psychological effects. O’Donnell and Bogenschutz [21,23], both from the University of New Mexico, centered on alcohol use, cravings, and psychological impacts. Agin-Liebes explored self-compassion and emotional regulation [22], while Garcia-Romeu and Johnson examined smoking cessation biomarkers and mystical experiences [26,28]. Dakwar assessed cocaine self-administration alongside mystical experiences, while Thomas and Loizaga-Velder concentrated on mindfulness, empowerment, and personal transformation, particularly in culturally integrated settings [25,27].

The settings varied from clinical environments to community-based retreats. O’Donnell, Agin-Liebes, and Bogenschutz conducted studies at the University of New Mexico [22,23] using structured environments conducive to psychotherapy. Glue evaluated the safety, tolerability, pharmacokinetics, and pharmacodynamics of single ascending doses of noribogaine in opioid-dependent patients seeking to discontinue methadone treatment, highlighting findings related to dose-dependent effects on QTc interval, opioid withdrawal symptoms, and adverse events [24]. Loizaga-Velder’s work was set in a traditional ayahuasca retreat in Mexico, emphasizing spiritual transformation, while Thomas conducted his research in a First Nations community with traditional ceremonies [25,27]. Garcia-Romeu and Johnson worked at Johns Hopkins University, and Dakwar operated out of Columbia University, focusing on cocaine addiction with ketamine [26,28,29,30].

These studies demonstrated that psychedelics have the potential to reduce substance use and enhance psychological well-being. O’Donnell and Bogenschutz found that psilocybin significantly reduced alcohol cravings [21,23]. Agin-Liebes reported that psilocybin helped with emotional release and improved coping, resulting in decreased alcohol consumption [22]. Garcia-Romeu and Johnson’s studies on smoking cessation showed long-term success, with up to 67% abstinence at 12 months [26,28,30]. Thomas and Loizaga-Velder reported positive effects on mindfulness and empowerment from ayahuasca therapy [25,27]. Dakwar’s research revealed that mystical experiences with ketamine reduced cocaine use, highlighting the role of subjective experiences in treatment [29].

The treatment approaches varied widely. O’Donnell and Bogenschutz combined manualized psychotherapy with psilocybin sessions, with O’Donnell using a 12-week program and scheduled psilocybin doses [21,23]. Agin-Liebes offered 8 h psilocybin sessions, while Garcia-Romeu and Johnson integrated psilocybin with cognitive behavioral therapy (CBT) and mindfulness techniques for smoking cessation [22,26]. Thomas and Loizaga-Velder used ayahuasca group therapy with a solid cultural emphasis [25,27]. Dakwar’s study was distinctive in using ketamine without psychotherapy, relying solely on the mystical experience for cocaine reduction [29].

Clinical settings like those used by O’Donnell, Agin-Liebes, and Bogenschutz emphasized structured environments with monitored psilocybin sessions and psychotherapy, showing evident reductions in alcohol use [21,22,23]. Garcia-Romeu and Johnson’s studies highlighted psilocybin’s effectiveness when combined with CBT for smoking cessation. In contrast, Thomas and Loizaga-Velder’s naturalistic, culturally embedded studies emphasized the spiritual and transformative potential of psychedelics [25,26,27,28,30]. Dakwar’s ketamine study, which lacked psychotherapy, demonstrated that psychedelic experiences alone can influence substance use outcomes, mainly through mystical experiences [29] (Table 2).

## 4. Discussion

Psychedelics have been used for thousands of years, primarily to heal both the body and mind, in shamanic rituals and for religious purposes [23,32]. One of the earliest instances of using psychedelics to treat alcohol addiction dates back to 1934, when Bill Wilson attempted to overcome his alcohol dependency through an experimental therapy involving a mixture of plants, with effects similar to psychedelics. Wilson experienced a “brilliant white light during this treatment”, which he described as a spiritual awakening and a moment of self-transcendence. Following this experience, he never drank alcohol again and later founded an international support organization for alcohol addicts [12].

Following the accidental discovery of LSD’s psychoactive properties by Albert Hofmann in 1943 [33] and the synthesis of psilocybin in 1958 [34], there was a significant increase in research on classic hallucinogens from the 1950s to the early 1970s. During that time, scientists studied the potential of these substances to treat alcohol and drug addiction, as well as conditions like anxiety, depression, and obsessive–compulsive disorder. Until 1970, LSD, psilocybin, and other hallucinogens were legally available for experimental research. More than a thousand publications from that era report on the treatment of over 40,000 individuals with classic hallucinogens for various psychiatric disorders [12,35,36].

After the 1970s, the line of research on PAT faced significant obstacles. The cultural and political climate shifted dramatically due to the widespread recreational use of psychedelics, which became associated with the counterculture movement [37]. Concerns over public health and safety led to strict regulatory actions. The Controlled Substances Act of 1970 classified LSD and psilocybin as Schedule I substances, indicating a high potential for abuse and no accepted medical use, and the United Nations Convention on Psychotropic Substances of 1971 placed psychedelics under control, leading many countries to adopt prohibitive regulations. This classification effectively halted clinical research by imposing stringent restrictions on possession, production, and clinical studies [38].

Classical psychedelics, such as psilocybin and LSD, primarily exert their effects through agonism at the serotonin 5-HT2A receptor, which is a well-established mechanism that underlies their psychoactive properties [39,40]. This receptor activation is associated with a range of neuroplastic changes, including promoting brain-derived neurotrophic factor (BDNF) signaling, which facilitates synaptic plasticity and neuronal growth [41,42]. In contrast, non-classical psychedelics, which include substances like MDMA and ketamine, may engage different receptor systems and mechanisms, leading to distinct psychoactive effects and therapeutic profiles [43,44].

Research indicates that psychedelics can be particularly effective in treating conditions like treatment-resistant depression, anxiety disorders, and substance use disorders. Johnson et al. emphasize the need to understand the mechanisms of action of psilocybin and related compounds, which can lead to long-lasting positive changes in mood and well-being, especially in individuals who have not responded to conventional treatments [6]. Furthermore, studies have shown that psychedelics do not lead to dependence or compulsive use, which is a significant advantage over many traditional psychiatric medications [40]. This is particularly relevant given the rising concerns about the addictive potential of various pharmacological treatments. As shown in Table 2, psychedelics contribute to the reduction of various symptoms [Table 2].

Psilocybin has garnered significant attention for its potential in treating mental health disorders, particularly depression. In recognition of its therapeutic promise, the U.S. Food and Drug Administration (FDA) granted a psilocybin “Breakthrough Therapy” designation for treatment-resistant depression in 2018 and for major depressive disorder in 2019 [45,46].

In recent years, there has been a resurgence of interest in the therapeutic potential of psychedelics for psychiatric disorders. The legal framework for conducting medical research with psychedelics varies by country but generally involves strict regulatory oversight. In the United States, researchers must obtain approval from the Food and Drug Administration (FDA) and a Drug Enforcement Administration (DEA) license to study Schedule I substances [47]. Internationally, the International Narcotics Control Board (INCB) monitors the implementation of UN drug control conventions, allowing for scientific and medical use under strict regulations [48].

### 4.1. Psychedelics Used in the Contemporary Era in the Treatment of SUD

#### 4.1.1. Psilocybin

Psilocybin (4-phosphoryloxy-N,N-dimethyltryptamine) is a well-known psychedelic compound primarily found in *Psilocybe mexicana* and other species of so-called “magic mushrooms” [1,49]. Historically, it was used by indigenous American cultures for religious, ritualistic, and healing purposes. Psilocybin, an indole alkaloid, metabolizes into psilocin and functions as a serotonin agonist. Over the past decade, interest in this substance has grown significantly. Research indicates that psilocybin is generally safe when administered in low doses within controlled settings. According to a review by Tylš et al. [50], low doses of psilocybin enhance mood, moderate doses induce a well-regulated altered state of consciousness, and high doses can lead to intense psychedelic experiences. Recent studies suggest that psilocybin holds potential for treating mental health conditions like depression, anxiety, obsessive–compulsive disorder, and SUD. Its effects on treating SUD, particularly in relation to alcohol and tobacco addiction, have been explored in recent evaluations [51,52].

#### 4.1.2. Lysergic Acid Diethylamide (LSD)

LSD acts as a partial agonist on 5-HT2/5-HT1 and dopaminergic receptors, and is initially synthesized as a potential stimulant for circulation and respiration [26,53,54]. Its commercialization by Sandoz laboratories allowed for numerous experimental studies, and it gained popularity as a recreational drug in the 1960s [33]. LSD is a potent substance that can induce significant changes in consciousness at doses as low as 20 μg, with fast absorption through the digestive system and primary metabolization in the liver. Research on LSD’s effects on addiction has mainly focused on alcoholism, with studies dating back to 1953, when psychiatrists Osmond and Hoffer introduced the drug to chronic alcohol users, exploring its potential to help individuals reach enhanced self-awareness [55].

#### 4.1.3. Ketamine

Ketamine, an arylcyclohexylamine with two enantiomers, (R)-ketamine and (S)-ketamine, was developed as a safer and more reliable alternative to phencyclidine for anesthesia [56]. It offers analgesic solid effects and is a short-acting anesthetic with minimal side effects. Ketamine is lipid-soluble and is commonly administered via injection, either intravenously or intramuscularly, taking effect on the central nervous system within 5 min [57]. Despite its medical applications, ketamine carries a significant risk of abuse and addiction. Doses between 1 and 4.5 mg/kg are typically used for anesthesia induction, with smaller doses for maintenance, and subanesthetic doses are used for psychedelic experiences [58]. Research has demonstrated ketamine’s effectiveness in treating major depressive disorder, including treatment-resistant cases, and in reducing suicidal thoughts. A single dose can produce fast and lasting effects for up to two weeks, and repeated low-dose infusions have proven even more beneficial in treatment-resistant depression [59]. Overall, studies have investigated the effect of ketamine in SUDs related to alcohol, cocaine, and heroin for the management of withdrawal symptoms.

#### 4.1.4. Ayahuasca

Ayahuasca is a psychoactive brew containing DMT and beta-carbolines, which activate DMT in the mouth. This combination produces longer-lasting altered states of consciousness (up to 6 h or more) with intense visual and emotional experiences. Beta-carbolines’ presence in ayahuasca also enhances its neurobiological effects [60]. It has been traditionally used by indigenous communities in the Amazon Basin for centuries and holds significant religious importance for groups like União do Vegetal and Santo Daime [61]. Many participants in these rituals have reported overcoming substance and alcohol addiction. A 2015 study found that ayahuasca effectively blocked the development and reinstatement of behavioral sensitization to alcohol in mice [62].

Cross-sectional research consistently shows that members of religious sects in Brazil and the U.S. who use ayahuasca have lower rates of alcohol abuse. Fabregas’s assessment of ayahuasca users indicated that they scored lower on the Addiction Severity Index (ASI) for alcohol use and psychiatric conditions compared to a control group [63,64].

Today, ayahuasca is being used in treatment centers and in both traditional and “neo-shamanic” settings to address various conditions, such as addiction and PTSD, as well as for purposes of personal and spiritual growth.

#### 4.1.5. Ibogaine

Ibogaine is a psychoactive alkaloid derived from the root bark of *Tabernanthe iboga*, a shrub native to West Africa [65]. Traditionally used in rituals by the Bwiti religion, ibogaine has psychedelic properties and is reported to reduce opioid withdrawal symptoms and cravings [66]. Its mechanism of action is not fully understood, but involves rapid conversion into its active metabolite, noribogaine, which has a longer elimination half-life. Both ibogaine and noribogaine interact with various neural pathways, potentially modulating opioid receptors and other systems [67]. The effects of ibogaine include reduced withdrawal symptoms, light sensitivity, and altered perception, though it also carries risks such as QT interval prolongation, which necessitates careful monitoring during use [66,67].

Clinical observations indicate that ibogaine can significantly reduce the acute symptoms of opioid withdrawal, as well as cravings associated with substance use disorders. Studies have shown that individuals undergoing ibogaine treatment often report a rapid decrease in withdrawal symptoms, as measured by standardized scales such as the Subjective Opiate Withdrawal Scale (SOWS) [68,69]. For instance, a twelve-month follow-up observational study highlighted a sustained reduction in opioid use and cravings among participants after ibogaine treatment, suggesting long-term benefits beyond the immediate detoxification phase [70]. Furthermore, ibogaine’s efficacy is often linked to the profound psychological experiences it induces, which may facilitate introspection and emotional processing, thereby contributing to recovery [71].

### 4.2. Psychedelic Therapy in the Last Ten Years of Research

Our systematic review found five relevant studies in the previous ten years that investigated the effect of psychedelics on patients with alcohol dependence using different substances (psilocybin and ayahuasca brew).

Psilocybin has a relatively short duration of consciousness-altering effects (4–6 h) and a well-established safety profile, especially in clinical settings [21,23]. It has been shown to promote neuroplasticity, which may help with sustained behavior change, and its therapeutic effects are often tied to the intensity of the psychedelic experience, including mystical-type experiences that shift participants’ values and self-identity [22].

These newest studies emphasize that these substances have the potential to catalyze behavioral and psychological changes, particularly by addressing emotional trauma and increasing self-awareness [21,23,25]. Especially, Bogenschutz et al.’s studies have demonstrated reductions in heavy drinking days and improved self-efficacy in managing alcohol cravings. Importantly, psilocybin’s ability to induce “quantum” change experiences—sudden, profound shifts in the psychological state—is critical to its effectiveness [23]. These studies support earlier discoveries from the peak period of psychedelic research, which suggest that psilocybin can significantly reduce alcohol consumption and cravings, improve mental health, and promote overall well-being [10,34,55,72,73]. Key results include long-term reductions in alcohol use and enhanced psychological outcomes. Despite these promising findings, psilocybin is not widely implemented in practice due to regulatory challenges, a lack of large-scale clinical trials, and more research to establish its long-term safety and effectiveness. Furthermore, societal stigma and legal obstacles around psychedelics limit their adoption in mainstream medical treatments.

Across the studies, participants commonly reported improved self-compassion and emotional regulation. Both psilocybin and ayahuasca are believed to help individuals confront past trauma, negative emotions, and self-critical thoughts, which are often associated with addiction [21,25,74]. Psilocybin-treated participants had about 41% fewer heavy drinking days compared to those receiving a placebo [22]. On the other hand, Thomas et al. report statistically significant improvements in mindfulness and emotional regulation, and reductions in problematic substance use (e.g., cocaine) in people who use ayahuasca as a therapy. However, some studies report varying effects on different substances, with reductions in alcohol and cocaine use, but not in cannabis or opiates [31,75].

All alcohol dependence studies included highlight the importance of set and setting—whether through guided therapy sessions in clinical settings or through the ritualized use of substances like ayahuasca in culturally traditional settings [25,27]. The supportive context, whether religious or therapeutic, enhances the effectiveness of the psychedelics. The follow-up periods in these studies often extended to several months, and participants continued to show reductions in heavy drinking days.

Looking for studies about smoking cessation and tobacco dependence, we discovered, in the last ten years, four original papers that studied the effect of psychedelics in the process of quitting smoking. Specifically, psilocybin was frequently used in the treatment of tobacco dependence with promising results, revealing both common patterns and critical differences in the outcomes and methodologies.

Across the studies analyzed, psilocybin demonstrated a substantial capacity to promote smoking cessation. In an open-label pilot study, Johnson et al. in 2014 reported that 80% of participants (12 out of 15) achieved biologically confirmed smoking abstinence at six months post-treatment [26,28]. This high success rate far exceeds the 35% success rates typically observed with other pharmacological and behavioral interventions for smoking cessation [28,30]. Similar results were observed in long-term follow-ups, where 60% of participants remained abstinent after 30 months (Noorani et al.) in 2018 [31].

These findings consistently highlight the potential of psilocybin as an effective adjunct to CBT in smoking cessation treatments [76,77]. In all studies, the psychedelic experience was framed within a structured therapeutic program that included preparatory counseling, mindfulness training, and post-session integration discussions [26,31]. This combination of psilocybin with CBT appears to be essential in sustaining therapeutic gains, as participants were encouraged to process their psychedelic experiences in ways that supported long-term abstinence from smoking. These studies consistently report that the preparatory sessions helped participants frame their motivations for quitting, while the follow-up sessions reinforced their abstinence [28,31].

Differences across studies include the variability in the dosage and number of psilocybin sessions administered. In some studies, participants received two doses of psilocybin (20 mg/70 kg and 30 mg/70 kg) spaced two weeks apart, with an optional third high-dose session [26,28]. In others, the dose and the number of sessions were modified based on participants’ smoking behaviors post-treatment [30]. While all participants underwent at least one high-dose session (typically 30 mg/70 kg), additional doses were tailored to individual needs, particularly for those who struggled to maintain abstinence. This flexible dosing strategy may contribute to the success rates observed across different follow-up periods. The duration of follow-up also varied between studies. While most studies conducted follow-ups at six months, some extended their follow-up periods to over 30 months, revealing that the effects of psilocybin on smoking cessation could persist for years [28]. Interestingly, while the success rate at six months was higher (80%), it decreased slightly over time, with 60% of participants remaining abstinent at long-term follow-up [30]. This decrease suggests that while psilocybin facilitates significant initial behavior change, maintaining long-term abstinence may require additional support or booster sessions.

Psilocybin’s ability to evoke profound changes in self-concept and emotional processing appears to be crucial in promoting smoking cessation. Participants across all studies reported that their psilocybin sessions provided vivid insights into their reasons for smoking, which often resulted in a re-evaluation of their identity and behavior [26,30,31]. A common theme reported by participants was the realization that smoking no longer made sense in the context of their newly discovered sense of self. This shift in perspective, coupled with the supportive therapeutic framework, allowed participants to break free from their addiction to cigarettes. Additionally, in concordance with older studies, many participants noted that the acute effects of psilocybin, including heightened emotional awareness and introspection, overshadowed any withdrawal symptoms they might have experienced, making it easier to abstain from smoking [77,78].

The precise psychological and neurobiological mechanisms by which psilocybin facilitates smoking cessation remain unclear. While mystical experiences are consistently associated with successful outcomes, more research is needed to determine how these experiences translate into lasting behavior change. Neuroimaging studies and longitudinal assessments of cognitive and emotional changes following psilocybin treatment could provide valuable insights into the underlying mechanisms of action.

A 2016 New Zealand study investigated the safety and efficacy of noribogaine, a metabolite of ibogaine, in opioid-dependent patients undergoing methadone detoxification. The study found that noribogaine was generally well tolerated but caused a dose-dependent prolongation of the QTc interval, a measure of heart rhythm. There was a non-statistically significant trend toward reduced opioid withdrawal symptoms, particularly at the 120 mg dose. Still, the time to resumption of opioid substitution therapy (OST) was not significantly different between the placebo and treatment groups [24].

Looking to the future, the clinical trial NCT04003948, titled “Preliminary Efficacy and Safety of Ibogaine in the Treatment of Methadone Detoxification”, is a Phase 2 study designed to evaluate the safety and efficacy of ibogaine hydrochloride in facilitating methadone detoxification. Similarly, the clinical trial NCT03380728, titled “Ibogaine in the Treatment of Alcoholism: a Randomized, Double-blind, Placebo-controlled, Escalating-dose, Phase 2 Trial”, aims to assess the safety and efficacy of ibogaine hydrochloride in treating alcohol dependence.

While the New Zealand study provides preliminary insights into the potential of noribogaine for opioid detoxification, the ongoing clinical trials NCT04003948 and NCT03380728 are expected to offer more comprehensive data on the safety and efficacy of ibogaine hydrochloride in treating substance dependence. The outcomes of these trials will be crucial in determining the therapeutic viability of ibogaine and its derivatives for addiction treatment.

The treatment of illicit drug dependence has seen growing interest in the integration of psychedelics and psychoactive substances, such as ayahuasca, ketamine, and psilocybin. Each study emphasizes the importance of the psychoactive effects of the substances used—whether through mystical-type experiences induced by ketamine or psilocybin or the visionary and emotional effects produced by ayahuasca. For instance, the ketamine study by Dakwar et al. demonstrated that the mystical-type experiences generated by ketamine were significantly associated with improvements in cocaine dependence [27,29,79].

Across the studies, there were substantial reductions in the use of substances following psychedelic or psychoactive treatment. In the ayahuasca-assisted therapy studies, participants reported reductions in or the complete cessation of problematic substance use, including cocaine, alcohol, and tobacco [27,79]. Likewise, in the ketamine study, which did not involve a cultural or spiritual context, participants demonstrated marked reductions in cocaine self-administration and cravings following treatment. However, ketamine’s ability to induce sustained behavior change appears to be more dependent on the intensity of the mystical-type experience, and the long-term effectiveness of ketamine without ongoing therapy or additional doses remains less understood compared to ayahuasca’s results [29].

### 4.3. The Future of Psychedelic Therapy

The future of psychedelic therapy in research and clinical practice for addictive use disorders holds significant promise, yet it necessitates rigorous investigation and cautious implementation. Recent studies have demonstrated that psychedelics such as psilocybin, ketamine, and ayahuasca may effectively reduce substance use, prolong periods of abstinence, and decrease the frequency of heavy usage days among individuals with SUD. These substances are believed to enhance neural plasticity, potentially reorganizing the disordered neural pathways involved in addiction. Additionally, psychedelics may attenuate maladaptive signaling in the mesolimbic reward pathway, offering a novel mechanism for treating SUD [80,81].

However, the existing research is limited by small sample sizes, methodological constraints, expectancy biases, and challenges with blinding procedures. The therapeutic effects observed are often partial and temporary, suggesting that repeated administrations and integration with psychotherapy might be necessary for enduring benefits. The combination of psychedelic treatment with psychotherapy appears to be more effective than either approach alone, highlighting the importance of a comprehensive therapeutic framework.

In research and therapeutic environments, administering classic hallucinogens requires careful safety measures to reduce risks. Participants must be screened thoroughly, excluding individuals with psychotic or bipolar disorders. A physician should be available during sessions, although the need for emergency interventions is rare. Psychological difficulties are typically addressed through reassurance from staff. Building a solid rapport between volunteers and staff before sessions helps alleviate anxiety and prepares individuals for the drug’s powerful effects. Volunteers are constantly monitored closely and never left alone during the experience. Afterward, follow-up meetings allow participants to reflect on their emotional or spiritual experiences and receive further care if necessary.

Future research should focus on conducting high-quality, controlled studies with more extensive and diverse populations to confirm the efficacy and safety of PAT. Longitudinal studies are required to determine the longevity of therapeutic effects, and neuroimaging research could elucidate the neural correlates of long-term changes induced by psychedelics.

Regulatory challenges persist, as many psychedelics are constrained by restrictive schedules that limit their medical use. A shift in regulatory status, along with the development of accredited training programs for clinicians, would be essential for integrating psychedelic therapies into mainstream clinical practice. Addressing these hurdles could pave the way for psychedelics to become a valuable addition to the repertoire of treatments for SUD, particularly for patients who have not responded to conventional interventions. Ultimately, while psychedelic therapy offers a promising avenue for the treatment of addictive use disorders, its future relies on thorough research, ethical considerations, and careful implementation to ensure safe and effective outcomes for patients.

## 5. Conclusions

Our research succeeded in reviewing the last ten years of international experience, highlighting the growing potential of PAT for treating SUD, especially in relation to alcohol, tobacco, and cocaine dependence. Recent studies demonstrate that psychedelics, such as psilocybin, ayahuasca, and ketamine, can significantly reduce substance use and cravings and promote psychological well-being. These effects are often linked to mystical or emotional experiences induced during therapy, which can catalyze profound behavioral changes. The integration of psychedelics with psychotherapy is critical for maximizing therapeutic outcomes. Regulatory and societal barriers, along with the need for specialized clinical training, must be addressed to ensure the safe and effective implementation of psychedelic therapies in mainstream addiction treatment. As research progresses, psychedelics could offer an innovative approach to SUD, particularly for individuals who have not responded to conventional treatments.

## Figures and Tables

**Figure 1 medicina-61-00278-f001:**
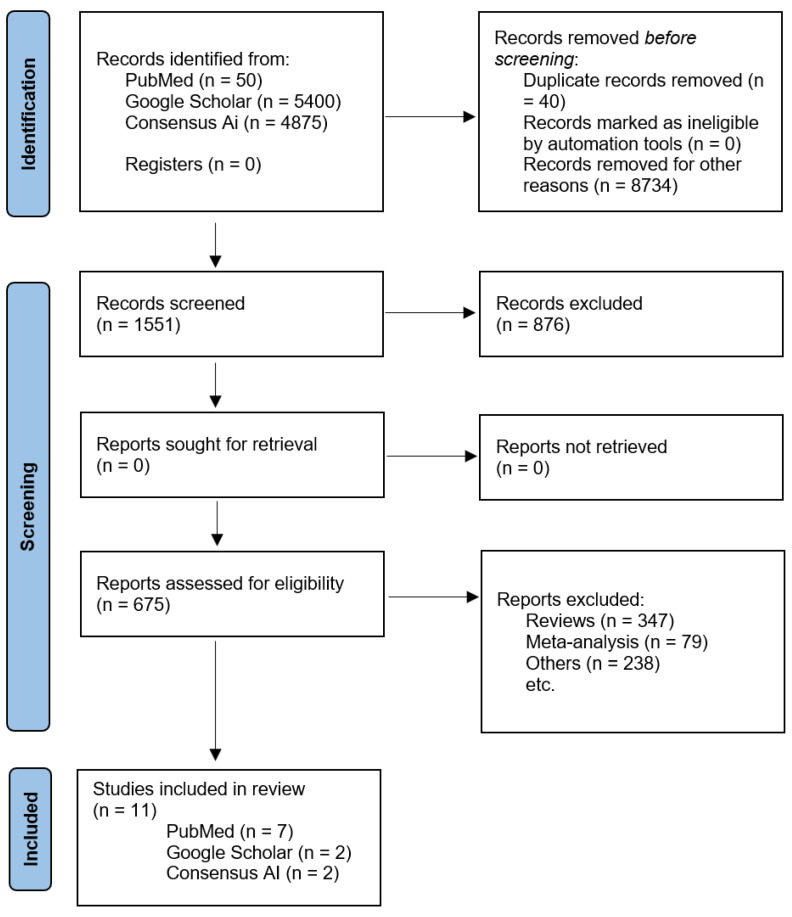
PRISMA flowchart of selected papers for study.

**Table 1 medicina-61-00278-t001:** Details of studies.

Name of First Author (Country)	Year of Publication	DOI	Design of the Study	Number of Participants	Type of Dependents Studied	Kind of Psychedelic Used and Dosage	Target Condition
K.C. O’ Donnell (USA) [21]	2022	10.1016/j.cct.2022.106976	RCT ^1^	96	Alcohol dependence	Psilocybin: 25 mg/70 kg	Treatment-seeking alcohol-dependent individuals
G. Agin-Liebes (USA) [22]	2023	10.1037/adb0000935	RCT ^1^	13	Alcohol dependence	Psilocybin: 25 mg/70 kg	A qualitative study to delineate psychological mechanisms of change in psilocybin-assisted therapy to treat alcohol addiction
M.P. Bogenschutz (USA) [23]	2015	10.1177/0269881114565144	OLPS ^2^	10	Alcohol dependence	Psilocybin, dosage not specified	Alcohol dependence
P. Glue (New Zealand) [24]	2016	10.1002/cpdd.254	RCT ^1^	27	Opioid dependence	Noribogaine: 60 mg, 120 mg, and 180 mg	Opioid withdrawal and cessation of methadone opioid substitution therapy
S. Loizaga-Velder (Mexico) [25]	2014	10.1080/02791072.2013.873157	Qualitative study	11	Alcohol dependence	Ayahuasca, traditional ritual doses	Participants with personal experience of ayahuasca rituals
A. Garcia-Romeu (USA) [26]	2015	10.2174/1874473708666150107121331	OLPS ^2^	15	Smoking cessation (tobacco)	Psilocybin: 20 mg/70 kg, 30 mg/70 kg	Tobacco addiction in long-term smokers
G. Thomas (USA) [27]	2013	10.2174/15733998113099990003	Prospective cohort study	12	Alcohol, tobacco, and cocaine	Ayahuasca, traditional ritual doses	Addiction, particularly problematic substance use, including alcohol and cocaine
M.W. Johnson (USA) [28]	2014	10.1177/0269881114548296	OLPS ^2^	15	Smoking cessation (tobacco)	Psilocybin: 20 mg and 30 mg/70 kg	Tobacco smokers
E. Dakwar (USA) [29]	2018	10.1016/j.neuropharm.2018.01.005	RCT ^1^	20	Cocaine use disorder	Ketamine: 0.71 mg/kg	Cocaine-dependent individuals
M.W. Johnson (USA) [30]	2017	10.3109/00952990.2016.1170135	Cross-sectional study	358	Smoking cessation (tobacco)	Psilocybin/LSD	Tobacco smokers
T. Noorani (USA) [31]	2018	10.1177/0269881118780612	Qualitative Study	30	Smoking cessation (tobacco)	Psilocybin: dosage not specified	Individuals who participated in psychedelic therapy

^1^ Randomized controlled trial. ^2^ Open-label pilot study.

**Table 2 medicina-61-00278-t002:** Psychedelic-Assisted Therapies for Substance Use Disorders and Behavioral Addictions—Key Studies and Findings.

Name of First Author (Country)	Measures	Setting	Main Findings	Treatment Program
K.C. O’Donnell (USA) [21]	Alcohol use (timeline follow-back), craving, self-efficacy	University of New Mexico	Reduction in heavy drinking days, with psilocybin showing promising results	12-week psychotherapy + psilocybin sessions at weeks 4 and 8
G. Agin-Liebes (USA) [22]	Self-compassion and affect regulation	Clinical setting	Psilocybin sessions promoted emotional release, reduced shame, improved coping with stress, and decreased alcohol cravings and consumption	8 h psilocybin administration sessions with a dose of 25 mg/70 kg of psilocybin or an active placebo (diphenhydramine)
M.P. Bogenschutz (USA) [23]	Alcohol use, craving, psychological effects	University of New Mexico	Significant reduction in alcohol craving and use	Manualized psychotherapy with psilocybin
P. Glue (New Zealand) [24]	Opioid withdrawal rating scales (COWS, OOWS, SOWS); pharmacokinetics and safety monitoring, including QTc interval measurements; pupil diameter measurements; time to the resumption of opioid substitution therapy (OST)	Zentech Clinical Trials Unit, Dunedin, New Zealand	Noribogaine was well tolerated but caused a dose- and concentration-dependent QTc interval prolongation. Noribogaine had a non-statistically significant trend toward reduced opioid withdrawal symptoms, particularly at the 120 mg dose. Time to OST resumption was not significantly different between placebo and treatment groups. Side effects included headache, nausea, and changes in light perception	Participants were transitioned from methadone to morphine before study treatment. Noribogaine or placebo was administered as a single oral dose, and participants were monitored for safety and pharmacodynamics over 35 days
S. Loizaga-Velder (Mexico) [25]	Personal narratives and qualitative reports	Ayahuasca Retreat in Mexico	Positive personal transformation linked to ayahuasca	Traditional ayahuasca retreat
A. Garcia-Romeu (USA) [26]	Smoking biomarkers, TLFB, QSU, SASE, HRS, SOCQ, Mysticism scale	Johns Hopkins University	Psilocybin treatment resulted in significant long-term smoking cessation success	Two to three psilocybin sessions (20 mg/70 kg and 30 mg/70 kg), integrated with 15 weeks of CBT and mindfulness-based techniques for smoking cessation
G. Thomas (USA) [27]	Philadelphia Mindfulness Scale (PHLMS), Empowerment Scale (ES), Hope Scale (HS), McGill Quality of Life (MQL), and 4-Week Substance Use Scale (4 WSUS)	Rural First Nations community with traditional longhouse-based ceremonies	Ayahuasca-assisted therapy improves mindfulness empowerment and reduces substance use	Ayahuasca-assisted group therapy combined with counseling for substance dependence
M.W. Johnson (USA) [28]	Smoking abstinence, CO, cotinine levels, Mystical Experience Questionnaire (MEQ)	Johns Hopkins University	67% abstinent at 12 months, 60% at long-term follow-up	Psilocybin with CBT
E. Dakwar (USA) [29]	Cocaine self-administration, Hood Mystical Experience Scale (HMS), dissociation	New York State Psychiatric Institute, Columbia University	Mystical-type experiences mediated reduction in cocaine use	Ketamine with no psychotherapy
M.W. Johnson (USA) [30]	Fagerström Test for Cigarette Dependence (FTCD), Questionnaire on Smoking Urges (QSU)	Non-laboratory environments	Psychedelics may facilitate long-term smoking cessation and reduce cravings	Psilocybin for tobacco cessation, integrated with CBT
T. Noorani (USA) [31]	Qualitative interviews	Various psychedelic therapy trials		PAT

## Data Availability

All data discussed in this review are available in the original publications cited in the reference list.

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
