# Peer review of "The Therapeutic Potential of Psychedelics in Treating Substance Use Disorders: A Review of Clinical Trials"

_medicina, 2025, doi:10.3390/medicina61020278_

Round 1
Reviewer 1 Report
Comments and Suggestions for Authors
This study aimed to systematically review current evidence on psychedelic-assisted therapy (PAT) for treating substance use disorder (SUD). The study focused on clinical trials and randomized controlled trials conducted from 2013 to 2023, and the key findings from 16 clinical trials investigating PAT for SUD were discussed. The topic itself is interesting, the manuscript is informative and generally well written. There are, however, several major concerns that need to be addressed.
The main concern is a lack of some important studies from the last ten years. Since the study focused on clinical trials, “clinicaltrials.gov” should have been one of the main sources of information. For example, ibogaine, derived from the African plant Tabernanthe iboga, has been investigated in the treatment of opioid addiction (and some other SUDs). There is a recent systematic literature review of clinical trials and therapeutic applications of ibogaine by Köck et al, and one of the clinical studies that should have been included in this manuscript is: Glue P, et al. Ascending single‐dose, double‐blind, placebo‐controlled safety study of noribogaine in opioid‐dependent patients. Clinical pharmacology in drug development. 2016;5(6):460-8. There are probably some other studies that were missed and should have been included. Besides, there are two current clinical trials for ibogaine that need to be discussed (as well as for other psychedelics): (1) https://clinicaltrials.gov/study/NCT04003948?intr=Ibogaine%20Hydrochloride&rank=1, (2) https://clinicaltrials.gov/study/NCT03380728?intr=Ibogaine%20Hydrochloride&rank=2
On the other hand, the inclusion of some studies is disputable. For example, in the study by E. Argento (Canada) [ref. 25], the treatment program was not described, and in my opinion it should be an exclusion criteria.
In Methodology, it needs to be clarified for the records removed before screening, what are “other reasons” (8734 records were removed this way)?
One of the major concerns is also that the authors stated as the aim to evaluate the mechanisms of both classic and non-classic psychedelics. However, it has not been covered in the manuscript. It was only stated that “psychedelics may attenuate maladaptive signaling in the mesolimbic reward pathway” (line 475), however without a reference.
The current role of psychedelics in treating a range of mental health conditions in general (besides SUD) should be discussed in the Introduction section, and the most promising results of PAT should be stated. For example, the studies on the use of psilocybin in depression treatment should be at least mentioned, as well as that psilocybin has been granted "Breakthrough Therapy" status by the FDA for depression treatment. I find it very important.
There are also some other concerns, as follows:
Title: It should be rephrased. It cannot be “A Review of the Last Decade”, but the review of clinical trials or achievement in PAT in the last decade.
Line 50: it refers to adult population, not general population.
Line 51: Specify the report (by WHO).
Lines 59-60: “according to 2019 studies” – please, clarify.
Line 508: “Our research succeeded in reviewing the last ten years of international experience” – it is not clear, what international experience?
The abbreviations need to be defined and used consistently. For example, PTSD is not defined in the text. PAT and SUD are used inconsistently throughout the text, either in its full or abbreviated form. The term “cognitive-behavioral therapy” (CBT) is also inconsistently stated throughout the text.
International non-proprietary names such as psilocybin and ayahuasca are globally recognized names used to identify the active ingredient in a medicine and have to be written with the first lowercase letters.
Author Response
Thank you for valuing our review and your involvement in improving the quality of the study.
We have analyzed the suggestion to include the article by Glue P. et al. in the review and deemed it important, therefore incorporating it into the studied articles. Additionally, we have reviewed the absence of treatment methodology in Argento's article and decided to exclude it from the review. Therefore, the changes are reflected in the article in Table 1, the 5th row, and the text in lines 15, 189-211, 238-242.
By including the study in which ibogaine was used as a psychedelic in the treatment of SUD, we added some general details about ibogaine to the text (lines 412-431). Additionally, we discussed the newly introduced study in comparison to ongoing studies (lines 527-546).
The reasons for excluding 8,734 records, noted in Figure 1 as "other reasons," have been clarified in the Methods section between lines 133-137.
At your suggestion, we have slightly modified the title of this review: The Therapeutic Potential of Psychedelics in Treating Substance Use Disorders: A Review of Clinical Trials.
In lines 42 and 43, we referred to the adult population and added the WHO report on addictions as a reference. Also, we correct the mistake in line 53. We reformulate the first sentence from the conclusion part (line 540)
We have addressed all the mentioned issues. Abbreviations are now clearly defined and consistently used throughout the text. For example, PTSD has been appropriately defined, and the terms PAT and SUD are now used consistently, either in full form or as abbreviations. Similarly, the term "cognitive-behavioral therapy" (CBT) has been standardized and used consistently across the document.
We have reviewed the aspects related to the mechanisms of classical and non-classical psychedelics between lines 344-351.
We have incorporated the requested content into the text. The current role of psychedelics in treating a broad range of mental health disorders, beyond SUD, has been discussed in the Introduction section. Additionally, we have highlighted the most promising results of PAT, including studies on the use of psilocybin in treating depression. Furthermore, we have mentioned that psilocybin has been granted "Breakthrough Therapy" status by the FDA for the treatment of depression. These additions can be found at lines 51-56 and 351-375.

Reviewer 2 Report
Comments and Suggestions for Authors
Addiction or substance use disorder (SUD) is a global public health interest that affects millions of people every day around the world.
The authors observed that symptoms of SUD may include compulsive consumption, failed efforts to quit using, cravings, social and worklife problems, tolerance, and withdrawal symptoms. The most recent data from 2022 showed that the global drug user population rose to 292 million, marking a 20% increase in the past decade. The authors observed that although drug consumption has risen, treatment access is still limited.
Even though psychedelics are often restricted, interest in their therapeutic application is increasing for the treatment of depression, anxiety,and substance use disorders.
The authors work very meticulously and precisely present the current state of knowledge on psychedelic-assisted therapy, covering various substances and methods of application used for patients suffering from substance use disorders, such as tobacco, alcohol, or other drugs.
The work proposed by the authors deserves to be published in the Medicina. Some minor issues are suggested to the authors.
Discution:
Line 306- 309 – it would be advisable to administer psilocybin in moderate doses, low doses
" Research indicates that psilocybin is generally safe when administered in moderate doses within controlled settings. According to a review by Tylš et al.[40], low doses of psilocybin enhance mood, moderate doses induce a well-regulated altered state of consciousness, and high doses can lead to intense psychedelic experiences".
References
Line 618 – item 36 “V1019.Pdf.” is not very understandable
Author Response
We sincerely thank you for your insightful and valuable comments on our article.
We appreciate your suggestion and have made the necessary modification to reflect the administration of psilocybin in low doses. (line 392)
We correct the reference and complete the title, year and location. (line 780)

Round 2
Reviewer 1 Report
Comments and Suggestions for Authors
The authors modified and improved most parts of the manuscript. My main concern is that ongoing clinical trials are not included, but only data published as journal articles. Since this is a review of clinical trials, I think it should have been included too. Some of the examples of ongoing trials are: ClinicalTrials.gov ID: NCT03380728, ClinicalTrials.gov ID: NCT04003948...
However, in my opinion, the Editor should decide if it is obligatory to include clinical trials in a new revised form of the manuscript or not.